# The presence of enteropathy in HIV infected children on antiretroviral therapy in Malawi

**Julia Blaauw**[1]* *, **Jessica Chikwana**[2]☯, **David Chaima**[2], **Stanley Khoswe**[2], **Lyson Samikwa**[2], **Isabelle de Vries**[1], **Wieger Voskuijl**[1]

1 Department of Global Child Health, Amsterdam Institute for Global Health and Development, Amsterdam University Medical Centers, Amsterdam, The Netherlands, 2 Department of Pediatrics & Child Health, The Kamuzu University of Health Sciences, Blantyre, Malawi

☯ These authors contributed equally to this work.
* Julia.blaauw@hotmail.com

**Data Availability Statement:** Data sharing requests can be directed to dr W.P. Voskuijl at w.p.voskuijl@amsterdamumc.nl or to the secretariat of The Amsterdam Institute for Global Health and

## Abstract

### Background

Undernutrition and malnutrition in children in low- and middle-income countries contribute to high mortality rates. Stunting, a prevalent form of malnutrition, is associated with educational and productivity losses. Environmental enteric dysfunction (EED) and human immunodeficiency virus (HIV) infection worsen these conditions. This study seeks to investigate the presence of enteropathy using EED fecal biomarkers in HIV-infected children who are stable on antiretroviral therapy (ART) across various nutritional statuses. By understanding the interplay between EED, HIV, and nutritional status, this study aims to provide insights that can inform targeted interventions to optimize nutritional outcomes in HIV infected children.

### Methods/Principal findings

This study evaluated the levels of alpha-1-antitrypsin, calprotectin and myeloperoxidase in frozen fecal samples from 61 HIV infected (mean age 9.16 ±3.08 years) and 31 HIV uninfected (6.65 ±3.41 years) children in Malawi. Anthropometric measurements and clinical data were collected. The height-for-age z-score (-1.66 vs -1.27, p = 0.040) and BMI-for-age z-score (-0.36 vs 0.01, p = 0.037) were lower in HIV infected children. Enzyme-linked immunosorbent assays were used to measure biomarker concentrations. Statistical tests were applied to compare biomarker levels based on HIV status and anthropometric parameters. Myeloperoxidase, alpha-1-antitrypsin, and calprotectin concentrations did not differ between HIV infected and HIV uninfected children of different age groups. In HIV infected children from 5–15 years, there is no difference in biomarker concentration between the stunted and non-stunted groups.

### Conclusion/Significance

Our study found a higher prevalence of stunting in HIV infected children compared to uninfected children, but no significant differences in biomarker concentrations. This suggests no causal relationship between enteropathy and stunting in HIV infected children. These results

Development at secretariat@aighd.org. Requests will be evaluated based on their scientific merit, the integrity of the research purpose, and the protection of participant confidentiality. Access to the data will be provided in compliance with applicable privacy and ethical standards.

**Funding:** This research project was financially supported with 3500 euros by "Doctors for Malawi" (https://www.doctorsformalawi.org). The author J. Blaauw was personally supported by the "MC de Visser" fund with 2000 euros for personal expenses for the purpose of staying in Malawi during the conduct of the research (https://www.mcdevisserfonds.org/aanvragen.html). The funders had no role in study design, data collection and analysis, decision to publish, or preparation of the manuscript.

**Competing interests:** The authors have declared that no competing interests exist.

contribute to the understanding of growth impairment in HIV infected children and emphasize the need for further research, particularly a longitudinal, biopsy-controlled study.

# Introduction

Undernutrition or malnutrition is a public health problem in children in low- and middle-income countries (LMIC) and contributes to 45% of childhood mortality [1]. It includes wasting, defined as a weight-for-height z-score <-2 (WHZ), and stunting, a height-for-age z-score <-2 (HAZ) [1,2]. Malnutrition in the context of human immunodeficiency virus (HIV) infection causes even higher morbidity and mortality. Stunting, associated with lower levels of education and a loss of productivity (5–7) is the most common manifestation of malnutrition, affecting 22% of children under the age of 5 in 2020 [1,3].

The persistently high stunting rates are associated with environmental enteric dysfunction (EED) [4–7]. EED is a chronic condition affecting the small intestine, prevalent in populations facing unsanitary living conditions and poor hygiene. It is thought to be caused by repetitive exposure of the gut to fecal or human pathogens, mycotoxins, and nutrient deficits [8]. Histologically, EED is characterized by alterations in the intestinal mucosa, including villous blunting, crypt hyperplasia, and chronic inflammation [7]. EED may explain the limited effectiveness of nutritional supplementation in improving growth [4,9,10].

The gold standard for diagnosing EED is endoscopic gut biopsy, which is rarely feasible in young children in the majority of LMICs. Numerous biomarkers of EED measured in urine, blood, and stool have been proposed [11], including alpha-1-Antitrypsin (AAT) reflecting increased intestinal permeability and protein loss [12,13], calprotectin (CP) and myeloperoxidase (MPO) both reflective of the presence of intestinal inflammation [4,7,13,14].

Nutritional deficiencies are widespread in HIV-infected children and play a significant role in the progression of the disease and the impairment of children's growth and development [15]. HIV infection causes impaired long-term gastrointestinal (GI) mucosal integrity and permeability, resulting in increased bacterial translocation and immune activation [16]. HIV-enteropathy is characterized clinically by chronic diarrhea and malabsorption, resulting in growth impairment [4,17–19]. A better understanding of the extent to which enteropathy plays a role and the mechanism by which it causes growth retardation in HIV-infected children is required, as many children do not recover from the chronic diarrhea-malnutrition cycle, particularly in the context of HIV infection [4,20,21].

This study seeks to investigate the presence of enteropathy using EED fecal biomarkers in HIV-infected children who are stable on antiretroviral therapy (ART) across various nutritional statuses. By understanding the interplay between EED, HIV, and nutritional status, this study aims to provide insights that can inform targeted interventions to reduce the adverse impacts of these conditions on child health and development.

# Materials and methods

## Study design and participants

This study is a continuation of a previous cross-sectional pilot study conducted at the Paediatric Department of Queen Elizabeth Central Hospital (QECH), Blantyre, Malawi between November 2018 and February 2019. This was a single-center pilot study to determine the prevalence of wasting and stunting in HIV reactive children on ART. We have integrated anthropometry data from the previous study with our laboratory findings on fecal biomarkers.

Children between the age of 2 and 15, who are HIV infected and have been using ART for at least 6 months have been enrolled in this study during their regular clinic visit. HIV uninfected siblings, also between the ages of 2 and 15, were asked to participate as the control group. Children who were currently hospitalized, were receiving tuberculosis medication or were not accompanied by a guardian were excluded.

## Data collection

We collected both social and medical histories from all participants by using standard medical questions. HIV infected children, information about their HIV medical history was found in the electronic system of the clinic and in their health passports. This information included the type of ART the child was using at the time of the visit, the length of time the child had been using ART, the compliance with ART during the previous clinic visit, whether the child was receiving co-trimoxazole preventive therapy (CPT), whether a viral load was available (8,18–20) and when this sample was taken, and the World Health Organization (WHO) stage at the time of ART initiation. According to Malawian HIV treatment guidelines, the type of ART was classified as first or second line [22]. ART adherence is defined as good with an adherence of 95%-105% and bad with an adherence of < 95% or > 105%. When the last viral load was <1000 copies, it was classified as virally suppressed, and when $\geq$ 1000 copies were found, it was classified as virally not suppressed.

## Anthropometry

Anthropometry measurements were obtained by trained staff members. Children aged 0 to 5 years old were classified using WHO Anthro [23] and those aged 5 to 15 years old were measured using WHO AnthroPlus [24]. Wasting was defined as WHZ $\leq$ 2 standard deviations (SD) of the Growth Reference median, and stunting, HAZ $\leq$ 2 SD of the median. In the absence of reference data for WHZ in children over the age of 5 [24], wasting was defined as WHZ $\leq$ 2 SD for children under the age of 5 and BMI-for-age z-score (BAZ) $\leq$ 2 SD for children over the age of 5, as most comparable studies have done [25–27]. The mid upper arm circumference (MUAC) was measured in a subset of children.

## Ethical considerations

Ethical consideration for this study was obtained at COMREC (College of Medicine Research Ethics Committee, an independent scientific and ethics committee). Approval for this project has been granted by the Directors of the Queen Elizabeth Central Hospital and Umodzi Lighthouse clinic (HIV clinic). The department of Paediatrics and Child Health has approved this project as well. Parents and guardians of all participants have provided written, informed consent. All remaining samples will be destroyed as per COMREC recommendations after completion of the study.

## Enteropathy biomarker assessment

Anthropometry and questionnaire data were collected from 253 children. However, because collecting stool samples was more difficult logistically, sample collection stopped at a number of 114 children. Stool samples were collected from 3 subgroups; HIV infected with normal anthropometry, HIV infected with abnormal anthropometry and HIV uninfected siblings. At the day of collection, the samples were stored at -80˚C for analyses in the lab of the Childhood Acute Illness and Nutrition Network at the Biomedical Sciences Department at the College of Medicine, University of Malawi. They were stored there from the end of 2018 to the end of

2022, when the final analyses were performed. The delay is due to the Covid-19 pandemic, which made earlier analysis logistically impossible. We evaluated three enteropathy biomarkers (MPO, CP and AAT) using commercially available enzyme-linked immunosorbent assays (ELISA) kits (Immunodiagnostik AG, Germany). ELISA tests were performed as directed by the package insert. The final dilution of the biomarkers was determined by choosing the most appropriate concentration of a biomarker within the standard curve's linear range, using a 4-parameter algorithm. A dilution factor of 1:2500 was used for CP, 1:500 for MPO, and 1:25000 for AAT. For MPO, a cut-off level of 2000 ng/ml stool was used [28] and for AAT, 270 µg/g stool was used as cut-off [29]. The cut-off values for CP used in the literature vary with age. Because the average age of our participants will be high in comparison to other studies of EED biomarkers, we chose to use the suggested cut-off level for adults (50 µg/g), which has been shown in multiple studies to be suitable for children aged 4–17 years in LMICs [30].

## Statistical analyses

The data were analyzed using IBM SPSS Statistics (version 29). The variables were examined for normality. Outliers in continuous measurements were kept if they were legitimate outliers. Missing values were coded. A p-value of less than 0.05 was considered statistically significant. Baseline patient characteristics were assessed using descriptive statistics. Descriptive statistics are shown in the form of means and SD in the case of normally distributed variables, median and interquartile range (IQR) in the case of non-normally distributed data, or as numbers and % for categorical variables. The Mann Whitney U test was used to compare measurements at enrollment between infected and uninfected children for the abnormally distributed variables, 2 sample unpaired T-test was used for normally distributed variables. Chi-square tests were used to compare categorical variables between HIV infected and HIV uninfected children. Fisher's exact test was used for small samples. EED biomarker concentrations were used as continuous variables and are shown as medians and IQR. The Mann-Whitney-U test for continuous variables and the Chi-square test for categorical variables were used to compare biomarker concentrations between different groups based on HIV status or anthropometry. Age was used as a grouping variable to compare groups based on different age categories.

## Results

### Baseline characteristics

Characteristics of study participants are presented in Table 1. Stool samples of 114 children were successfully collected. 22 samples were unsuitable for further analysis due to ambiguity of origin, ambiguity of coding, or a lack of sufficient aliquots. 92 samples were used for analyses, with 61 (66.3%) HIV infected and 31 (33.7%) HIV uninfected children. 38.7% of the HIV uninfected children were male, compared to 57.4% of the HIV infected children. The mean age of the HIV infected children included in this study was 9.16 years (SD ±3.08), the mean age of the HIV uninfected children was 6.65 years (SD ±3.41), this was a statistically significant difference (p = <0.001). When participants were divided into various age groups, it was found that there was a statistically significant difference between HIV infected participants and HIV uninfected participants (p = 0.008).

**Anthropometric measurements.**   The height-for-age z-score (-1.66 vs -1.27, p = 0.040) and BMI-for-age z-score (-0.36 vs 0.01, p = 0.037) were statistically significantly lower in HIV infected children. The number of children with wasting or stunting did not differ between HIV infected and HIV uninfected. There was no difference in mid upper arm circumference (MUAC) between HIV infected and HIV uninfected children.

**Table 1. Baseline characteristics including anthropometric features from study participants.**

|  | HIV infected (n = 61) | HIV uninfected (n = 31) | P- value |
|---|---|---|---|
| **Gender (n, %)** |  |  | 0.090 |
| male | 35 (57.4%) | 12 (38.7%) |  |
| female | 26 (42.6%) | 19 (61.3%) |  |
| **Age in years (mean, SD)** | 9.16 (±3.08) | 6.65 (±3.41) | **<0.001** |
| **Age groups (n, %)** |  |  | **0.008** |
| 2–5 years | 7 (11.5%) | 12 (38.7%) |  |
| 6–10 years | 35 (57.4%) | 14 (45.2%) |  |
| 11–15 years | 19 (31.1%) | 5 (16.1%) |  |
| **Anthropometric measurements** |  |  |  |
| MUAC in cm (median, IQR) | N = 43<br>17.50 (16.40–19.00) | N = 23<br>17.10 (15.60–17.90) | 0.241 |
| height-for-age z-score (median, IQR) | -1.66 (-2.82 - -0.84) | -1.27 (-2.01 - -0.45) | **0.040** |
| weight-for-length z-score [a](median, IQR) | -0.03 (-1.45–0.53) | 0.43 (-0.48–1.46) | 0.497 |
| BMI-for-age z-score (median, IQR) | -0.36 (-0.95–0.13) | 0.01 (-0.50–0.88) | **0.037** |
| wasted (n, %) | 5 (8.2%) | 1 (3.2%) | 0.660 |
| stunted (n, %) | 25 (41.0%) | 8 (25.8%) | 0.151 |

Numbers in bold are statistically significant.

[a] weight-for-length z-scores were only calculated for children under the age of five.

**HIV medical characteristics.** The medical characteristics of the HIV infected children included in this study are shown in S1 Table. The mean duration on ART was 72.2 months (SD ± 38.3). 59.0% of the children were on ART for at least 60 months. Nearly 75% of all children were receiving first-line ART. The proportion of children with good or bad therapy adherence was roughly balanced. Every child was receiving co-trimoxazole preventive therapy. The median viral load was 839 (IQR 40–4964) copies. Over 60% of the children were virally suppressed.

## Prevalence of enteropathy in HIV infected and HIV uninfected children

To evaluate associations between enteropathy and HIV status, levels of fecal markers for enteropathy were compared between HIV infected and HIV uninfected children of different age groups. Biomarker concentrations between HIV infected and HIV uninfected children are shown in Table 2. MPO, AAT, and CP concentrations did not differ between HIV infected and HIV uninfected children of different age groups. There was no significant difference in the percentage of children with elevated levels (yes/no) between HIV infected and HIV uninfected children. A large proportion of children in both the HIV infected and HIV uninfected groups had elevated fecal levels of CP, especially the younger groups (2–5 years: 100.0% vs 91.7%, 6–10 years: 61.8% vs. 61.5%). S1 Fig shows boxplots of biomarker concentrations in HIV infected and HIV uninfected children. Because the age groups were too small for individual boxplots, boxplots of all children were created. The HIV infected group's data is more spread out, including several outliers in both categories.

**Association between enteropathy and stunting in HIV infected children.** Table 3 compares the concentrations of biomarkers and whether they were elevated (yes/no), between stunted and non-stunted children of different ages in the group of HIV infected children. MPO levels were statistically significant higher in the non-stunted group among the 2–5-year-old children. In HIV infected children from 5–15 years, there is no statistically significant

**Table 2. Concentrations of fecal enteropathy markers in HIV infected and HIV uninfected children of different age groups.**

| | HIV infected | HIV uninfected | p value |
|---|---|---|---|
| **Biomarker concentrations (median, IQR)** **2–5 years (n = 19)** | | | |
| MPO (ng/mL) | 1874 (458–7628) | 558 (468–2101) | 0.258 |
| AAT (μg/g) | 557 (6–2270) | 87 (13–427) | 0.574 |
| CP (μg/g) | 227 (141–624) | 92 (58–243) | 0.108 |
| Elevated MPO (n, %) | 4 (66.7%) | 3(27.3%) | 0.162 |
| Elevated AAT (n, %) | 4 (80.0%) | 4 (33.3%) | 0.131 |
| Elevated CP (n, %) | 7 (100.0%) | 11(91.7%) | 1.000 |
| **6–10 years (n = 49)** | | | |
| MPO (ng/mL) | 833 (399–1525) | 1348 (675–2189) | 0.471 |
| AAT (μg/g) | 205 (42–277) | 55 (26–176) | 0.051 |
| CP (μg/g) | 75 (20–180) | 84 (24–159) | 0.915 |
| Elevated MPO (n, %) | 6 (18.2%) | 3(21.4%) | 1.000 |
| Elevated AAT (n, %) | 11(33.3%) | 2(14.3%) | 0.288 |
| Elevated CP (n, %) | 21(61.8%) | 8(61.5%) | 1.000 |
| **11–15 years (n = 24)** | | | |
| MPO (ng/mL) | 328 (194–455) | 188 (140–624) | 0.233 |
| AAT (μg/g) | 71 (29–228) | 88 (27–115) | 0.766 |
| CP (μg/g) | 28 (6–83) | 38 (19–68) | 0.823 |
| Elevated MPO (n, %) | 3(15.8%) | 0 | 1.000 |
| Elevated AAT (n, %) | 3(16.7%) | 0 | 1.000 |
| Elevated CP (n, %) | 9(50.0%) | 1(20.0%) | 0.339 |

Cut-off values: MPO 2000 ng/ml, AAT 270 μg/g, CP 50 μg/g.

difference in biomarker concentrations between the stunted and non-stunted groups. There were statistically significantly more stunted children with increased AAT levels in the 6-10-year age range than non-stunted children.

**Association between enteropathy and stunting in HIV uninfected children.** S2 Table compares the concentrations of biomarkers between stunted and non-stunted children of different ages in the group of HIV uninfected children. In HIV uninfected children, there is no statistically significant difference in biomarker concentration between the stunted and non-stunted groups.

**Association between enteropathy and HIV status in stunted children.** S3 Table compares the concentrations of biomarkers between HIV infected and HIV uninfected stunted children of different ages. There was no statistically difference found in biomarker concentrations between HIV infected and HIV uninfected in the group of stunted children.

## Discussion

This study investigated the presence of enteropathy in HIV infected children with and without malnutrition. Stunting was more prevalent in HIV infected Malawian children, stable on ARTs, compared to the HIV uninfected group. However, fecal MPO, AAT, and CP levels did not differ between the HIV infected and uninfected group. We did not find a difference in biomarker concentrations between stunted and non-stunted HIV infected children. Additionally, there was no association between biomarker concentrations and stunting in the HIV uninfected group.

**Table 3. Concentrations of fecal enteropathy markers between stunted and non-stunted HIV infected children.**

|  | Stunted (n = 25) | Non-stunted (n = 36) | p value |
|---|---|---|---|
| **Biomarker concentrations (median, IQR)** |  |  |  |
| **2–5 years** |  |  |  |
| MPO (ng/mL) | 1163 (305–1874) | 12707 (2548–22868) | **0.034** |
| AAT (µg/g) | 557 (81–3878) | 699 (8–1390) | 1.000 |
| CP (µg/g) | 340 (183–962) | 138 (57–220) | 0.077 |
| Elevated MPO (n, %) | 1 (33.3%) | 3 (100.0%) | 0.400 |
| Elevated AAT (n, %) | 3 (100.0'%) | 1 (50.0%) | 0.400 |
| Elevated CP (n, %) | 4 (100.0%) | 3 (100.0%) |  |
|  |  |  |  |
| **6–10 years** |  |  |  |
| MPO (ng/mL) | 1466 (558–1975) | 778 (323–1274) | 0.182 |
| AAT (µg/g) | 275 (127–835) | 151 (42–238) | 0.124 |
| CP (µg/g) | 134 (38–180) | 67 (20–145) | 0.401 |
| Elevated MPO (n, %) | 2 (22.2%) | 4 (16.7%) | 1.000 |
| Elevated AAT (n, %) | 6 (66.7%) | 5 (20.8%) | **0.033** |
| Elevated CP (n, %) | 6 (66.7%) | 15 (60.0%) | 1.000 |
| **11–15 years** |  |  |  |
| MPO (ng/mL) | 372 (234–463) | 267 (188–404) | 0.821 |
| AAT (µg/g) | 51 (12–154) | 168 (54–295) | 0.134 |
| CP (µg/g) | 30 (4–80) | 28 (15–104) | 0.298 |
| Elevated MPO (n, %) | 1 (8.3%) | 2 (28.6%) | 0.523 |
| Elevated AAT (n, %) | 2 (16.7%) | 1 (16.7%) | 1.000 |
| Elevated CP (n, %) | 6 (54.5%) | 3 (42.9%) | 1.000 |

Cut-off values: MPO 2000 ng/ml, AAT 270 µg/g, CP 50 µg/g.

Children with HIV have a lower HAZ (-1.66 vs -1.27, p = 0.040) and BAZ (-0.36 vs 0.01, p = 0.037) compared to the HIV uninfected group. This finding is consistent with other studies; despite modest growth recovery after starting ART, relatively few kids achieve normal growth, partly due to the interacting effect of HIV infection and malnutrition [20,21,31,32]. A causal relationship between the presence of enteropathy and stunting could have partly explained why HIV infected children have poorer nutritional status than non-infected children, but no association between elevated biomarkers and stunting was found in the present study. We found higher MPO levels in younger children who were not stunted in comparison with the stunted group, contrary to our hypothesis. A possible explanation can be the heterogeneity of the groups in terms of socioeconomic, environmental and HIV related factors. Additionally, we discovered a higher percentage of increased AAT levels in HIV infected stunted children aged 6 to 10, indicating the presence of intestinal inflammation, consistent with our hypothesis.

In a recent systematic review [11], examining EED indicators to growth outcomes, many studies did not report participants' HIV status. No studies reported on the association between EED and stunting in HIV infected children. The review concluded that currently inconsistent hypotheses exist about the mechanism how EED leads to stunting. This implies that EED is a complex, possibly transient condition rather than a binary disorder (present or not) [11]. It is currently unknown to which degree HIV infection affects EED, consequently leading to stunting [4,20,21].

The biomarker concentrations found in this study are consistent with findings from other EED related studies in Malawi [33,34], although children included in those studies were younger and were HIV uninfected or their HIV status was unknown. The levels of MPO, AAT and CP were elevated in multiple children in this current study, irrespective of HIV status. This finding suggests some degree of intestinal inflammation and increased intestinal permeability in both HIV infected and uninfected children, consistent with the high prevalence of enteropathy in children in LMICs [4,35]. A possible explanation for this is that both groups of children are exposed to the same environmental factors. Furthermore, the HIV infected children did not have an advanced stage of HIV infection, since all HIV infected children received ART for at least 6 months and those who were in an advanced stage of the disease (admitted to the hospital or receiving tuberculosis treatment) were excluded. It is known that enteropathy-related gut changes are more commonly identified in patients with advanced stages of HIV infection [4]. Also, all HIV infected children were taking daily cotrimoxazole, which has been showed to reduce MPO levels [36].

This current study found that CP was especially high in younger children (2–5 years), which is consistent with previous studies that biomarker concentrations decrease with age [33].

So far, only a few studies investigated the presence of enteropathy in HIV infection, primarily in adults and using a different set of biomarkers [37,38]. In these studies, morphological and functional mucosal changes have been described in HIV infected patients, like a loss of villous architecture, increased permeability, mucosal CD4+ T cell depletion, leukocyte infiltration and microbial translocation [39–41].

The current study is not without limitations: firstly, we only measured biomarkers at one-time point without longitudinal data on growth or enteropathy. Secondly, children in the HIV infected group were older than the HIV uninfected. Children were divided into age groups to account for the age difference. However, finding a statistically significant difference among the smaller groups was more difficult. Thirdly, since it proved more difficult to collect fecal samples than we had anticipated, not all participants had complete data (both anthropometry and fecal sample).

In conclusion, our study showed a higher prevalence of stunting among HIV infected children compared to HIV uninfected children. However, we did not find significant differences in biomarker concentrations between the two groups, implying that there is no indication of a possible causal relationship between enteropathy and stunting in HIV infected children in the current study. To further investigate the mechanisms and role of enteropathy in poor growth in HIV infected children and, ultimately, to develop more specific interventions for this vulnerable group, a longitudinal, biopsy-controlled study will be preferred, considering affecting factors such as age, HIV therapy, and stage of disease.

## Supporting information

**S1 Fig. Boxplots showing concentrations of fecal enteropathy markers between HIV infected (blue) and HIV uninfected (orange).** Note. Y-axes report different units and scales.
(TIF)

**S1 Table. Medical history of HIV infected children.**
(TIF)

**S2 Table. Concentrations of fecal enteropathy markers between stunted and non-stunted HIV uninfected children.**
(TIF)

**S3 Table. Concentrations of fecal enteropathy markers between HIV infected and HIV uninfected stunted children.**
(TIF)

## Acknowledgments

We thank all of the children and their guardians who participated in the study, the Lighthouse clinic and its employees for their assistance and the laboratory of The Kamuzu University of Health Sciences' biomedical department.

## Author Contributions

**Conceptualization:** Jessica Chikwana, Wieger Voskuijl.

**Investigation:** Julia Blaauw, Jessica Chikwana, Isabelle de Vries.

**Resources:** Stanley Khoswe, Lyson Samikwa.

**Supervision:** David Chaima, Wieger Voskuijl.

**Validation:** David Chaima.

**Writing – original draft:** Julia Blaauw.

**Writing – review & editing:** Jessica Chikwana, Isabelle de Vries, Wieger Voskuijl.

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
