## [Decision Letter · Decision Letter 0]

19 Oct 2023

PONE-D-23-23008The presence of enteropathy in HIV infected children on antiretroviral therapy in MalawiPLOS ONE

Dear Dr. Blaauw,

Thank you for submitting your manuscript to PLOS ONE. After careful consideration, we feel that it has merit but does not fully meet PLOS ONE’s publication criteria as it currently stands. Therefore, we invite you to submit a revised version of the manuscript that addresses the points raised during the review process.

We look forward to receiving your revised manuscript.

Kind regards,

Furqan Kabir

Academic Editor

PLOS ONE

Journal Requirements:

4. Please upload a copy of Supporting Information Tables S1 to S3 which you refer to in your text on page 18.

Reviewers' comments:

Reviewer's Responses to Questions

**Comments to the Author**

1. Is the manuscript technically sound, and do the data support the conclusions?

Reviewer #1: Yes

Reviewer #2: Yes

2. Has the statistical analysis been performed appropriately and rigorously? 

Reviewer #1: Yes

Reviewer #2: Yes

3. Have the authors made all data underlying the findings in their manuscript fully available?

Reviewer #1: Yes

Reviewer #2: Yes

4. Is the manuscript presented in an intelligible fashion and written in standard English?

Reviewer #1: Yes

Reviewer #2: Yes

5. Review Comments to the Author

Reviewer #1: The authors conducted a well-designed study. The write up is clear and understandable. The authors discussed the finding and the limitations nicely. I have few comments to improve the clarity of the manuscript for the authors to consider.

•A brief introduction of EED with its histological features would be good.

•Why did the authors use these three specific biomarkers when Neopterin, LR, or LM ratios are also counted as very promising biomarkers of EED?

•“This study is a continuation of a previous cross-sectional pilot study”- please add few lines regarding that pilot study for the readers to understand the context well.

•Why did the sample collection stop at a number of 114 children? Please justify. Was any sample size calculated for this or it was just an arbitrary decision?

I think the use of ARTs were the main reason behind the non-significant results. The authors also discussed this.

Reviewer #2: This is an interesting study on the association between enteropathy and HIV infections among children in Malawi.

I propose to address the following issues.

Abstract:

Line 34: The author needs to mention the rationale of this study.

Introduction:

Line 78: The author needs to mention the rationale of this study. The need to state the specific knowledge gap in this field.

Methods:

Line 95: Is the questionnaire validated?

Need to declare the data sharing policy at the end of manuscript.

6. PLOS authors have the option to publish the peer review history of their article (what does this mean?). If published, this will include your full peer review and any attached files.

Reviewer #1: No

Reviewer #2: **Yes: **Md Kamruzzaman

---

## [Author Response · Author response to Decision Letter 0]

17 Jan 2024

Reviewer #1:

• Comment: reviewer #1 asked questions about why we used these biomarkers when Neopterin, LR of LM ratios are also counted as very promising biomarkers of EED?

Answer: Due to logistical and financial constraints, we were restricted to assessing only three biomarkers in our study. Neopterin, together with the two markers we used, CP and MPO, are indicative of intestinal inflammation. The decision to opt for CP and MPO over NEO was based on the comparable nature of their results and growth outcome evaluations. Given that we had already employed two biomarkers focusing on intestinal inflammation, we included AAT, assessing intestinal permeability, to present a more comprehensive perspective on intestinal functionality. While lactulose-mannitol (L:M) and lactulose-rhamnose (L:R) ratios are also reflective of permeability, their utilization is hampered by numerous limitations stemming from inconsistent associations with growth outcomes and the variability in L:M test results across different laboratories and platforms1,2. 

1. Denno et al. Clinical Infectious Diseases 2014;59:S213–9. 

2. Harper et al.Journal of XYZ 2018. 

• Comment: Why did the sample collection stop at a number of 114 children? Please justify. Was any sample size calculated for this or it was just an arbitrary decision?

Answer: Based on a power calculation, we aimed to recruit a total of 30 participants for each arm to detect a difference in primary outcome with α = 0.05 and 80% power, resulting in a total of 90 participants. Recruitment went very smoothly and faster than expected. Anthropometry and clinical data were collected from 253 children. However, because collecting stool samples proved to be more difficult logistically, we were only able to collect a stool sample of 114.

• Comment: A brief introduction of EED with its histological features would be good.

Answer: We have amended the introduction and added a few lines regarding the pathophysiology of EED to the manuscript. (please see page 3 line 67-73 of the revised manuscript)

• Comment: “This study is a continuation of a previous cross-sectional pilot study”- please add few lines regarding that pilot study for the readers to understand the context well.

Answer: An explanation of this is admitted in the manuscript. (please see page 4 line 97-100 of the revised manuscript)

Reviewer #2:

• Comment: Abstract: Line 34: The author needs to mention the rationale of this study. + Line 78: The author needs to mention the rationale of this study. The need to state the specific knowledge gap in this field.

Answer: We have revised the abstract and introduction section to properly reflect the study's rationale. (please see page 2 line 34-39 and page 4 line 84-92 of the revised manuscript)

• Comment: Methods:

Line 95: Is the questionnaire validated?

Answer: The questions in the ‘questionnaire’ are not validated but are very normal medical questions regarding housing, people in the family and questions regarding initiation of and compliance with ART’s. To prevent misunderstandings, we have removed the word or reference to a ‘questionnaire’ but framed it as ‘We collected both social and medical histories from all participants by using standard medical questions’. (please see page 5 line 106-107 of the revised manuscript)

• Comment: Need to declare the data sharing policy at the end of manuscript.

Answer: We recognize the importance of transparency and open access in scientific research and we are committed to prioritize the privacy and confidentiality of sensitive information. We believe this is the case in our manuscript given that the data includes details about the HIV status of vulnerable children and their families. When making this potentially stigmatizing and sensitive data/information publicly available, this can compromise the privacy and well-being of this vulnerable group. We would very much prefer not to make the data publicly available but upon request. We have added a non-author point of contact where data requests may be made. We very much hope this is possible and are looking forward to your response. 

We added to following statement to the manuscript (page 15): “Data sharing requests can be directed to dr W.P. Voskuijl at w.p.voskuijl@amsterdamumc.nl or to the secretariat of The Amsterdam Institute for Global Health and Development at secretariat@aighd.org. Requests will be evaluated based on their scientific merit, the integrity of the research purpose, and the protection of participant confidentiality. Access to the data will be provided in compliance with applicable privacy and ethical standards.”

---

## [Editor Report · Decision Letter 1]

23 Jan 2024

The presence of enteropathy in HIV infected children on antiretroviral therapy in Malawi

PONE-D-23-23008R1

Dear Dr. Blaauw,

We’re pleased to inform you that your manuscript has been judged scientifically suitable for publication and will be formally accepted for publication once it meets all outstanding technical requirements.

Kind regards,

Furqan Kabir

Academic Editor

PLOS ONE
---

## [Editor Report · Acceptance letter]

29 Jan 2024

PONE-D-23-23008R1 

PLOS ONE

Dear Dr. Blaauw, 

I'm pleased to inform you that your manuscript has been deemed suitable for publication in PLOS ONE. Congratulations! Your manuscript is now being handed over to our production team.

Kind regards, 

on behalf of

Dr. Furqan Kabir 

Academic Editor

PLOS ONE